# Social Governance and Sustainable Development in Elderly Services: Innovative Models, Strategies, and Stakeholder Perspectives

**Huaiyue Wang 1,2, Peter C. Coyte 2, Weiwei Shi 3 , Xu Zong 4 and Renyao Zhong 1,***

1   School of Public Administration, East China Normal University, Shanghai 200062, China;
    sww.wang@mail.utoronto.ca
2   Institute of Health Policy, Management and Evaluation, University of Toronto,
    Toronto, ON M5T 3M6, Canada; peter.coyte@utoronto.ca
3   School of Electronic Information and Electrical Engineering, Shanghai Jiao Tong University,
    Shanghai 200240, China; sirius.sww@gmail.com
4   Population Research Unit, Faculty of Social Sciences, University of Helsinki, 00014 Helsinki, Finland;
    xu.zong@helsinki.fi
*   Correspondence: zhangrenyao067@gmail.com

**Abstract:** Introduction: The global demographic shift towards an aging population has created an urgent need for high-quality elderly care services. This study focuses on "elder services" within the framework of sustainable development, addressing seniors with intensive care needs and independent seniors. Methods—Social Governance: To understand the social governance aspects, we employ a qualitative methodology, analyzing policy documents, novel care methods, and successful case studies. Sustainable Development: Simultaneously, our study investigates sustainable development, examining the methods used to promote sustainability in geriatric care. Research Question: Our research question centers on identifying strategies that foster inclusivity and sustainability in elder services, considering diverse needs, housing, community involvement, and the role of technology. Results: We identified innovative models aimed at improving the well-being of older individuals, including community-driven initiatives, technology-assisted solutions, holistic wellness programs, intergenerational interaction programs, and the integration of traditional and modern care methods. We explored stakeholder perspectives, providing insights into the complexities of implementing effective elderly care solutions. Our study evaluated the efficiency of diversified social governance models in geriatric care, highlighting their benefits compared to traditional models. We presented specific concerns and suggestions from stakeholders regarding sustainable development in geriatric care. Discussion: Our findings underscored the importance of collaboration among various stakeholders to enhance elderly care. Our study summarizes key insights from current policies and anticipated future trajectories in geriatric care, providing a foundation for developing sustainable elderly care facilities.

**Keywords:** elderly services; social governance; inclusivity; technology; collaboration; long-term strategies; ecological design; sustainable development

## 1. Introduction

The global demographic landscape is currently undergoing a significant shift, characterized by a substantial rise in the proportion of older people. According to the 2022 report from the United Nations (UN), the anticipated substantial increase in the share of adults aged 65 and above by the year 2050, from the current ratio of one in eleven to nearly one-sixth of the world population, underscores the growing imperative for social institutions to effectively respond to the evolving and varied needs of aging populations [1].

The current methods of caring for the elderly, which are often fragmented and insufficient, are proven to be insufficient due to changing societal dynamics and evolving

medical needs. There is an increasingly prevalent agreement among legislators, healthcare professionals, and community leaders regarding the imperative need for a more holistic approach to senior care. The effectiveness of different social governance strategies in providing geriatric services has emerged as a crucial factor in sustaining the welfare and general standard of living of the elderly population [2]. Nevertheless, it is important to note that the challenges faced by affluent and developing countries vary significantly, posing intricate problems for governments.

Within this environment, the notion of sustainable development assumes a pivotal position, urging global communities to actively pursue a state of equilibrium in the face of many challenges. This balance guarantees the prompt fulfillment of present needs while also safeguarding the long-term sustainability of proposed solutions from both environmental and economic perspectives. Understanding the intricate needs of older adults is of paramount importance in the field of geriatric care. The theoretical underpinning of sustainable development places significant emphasis on the integration of social, economic, and environmental variables [3]. This paradigm offers significant insights into the efficient management of geriatric services, thereby ensuring long-term social advantages [4].

This study's significance is grounded in its unique approach to the intersection of geriatric care and sustainable development, addressing pressing issues while highlighting the core problem areas: Firstly, the challenge of an aging global population calls for a reevaluation of conventional elderly care practices. The problem is the inadequacy of fragmented approaches to meeting the diverse needs of seniors, impacting their quality of life and societal well-being. Secondly, integrating sustainable development principles into geriatric care is essential to aligning with social justice, financial sustainability, and ecological conservation. The problem is the misalignment of elderly care practices with these principles, exacerbating environmental and economic challenges. Thirdly, optimizing geriatric care programs to efficiently meet rising demands is imperative. The problem is the inefficiency and ineffectiveness of care delivery, resulting in reduced quality of life for seniors and straining healthcare systems.

In response to these challenges, the study aims to explore innovative models, strategies, and stakeholder perspectives to arrive at a plan for social governance and sustainable development in elderly services. This study presents novel research opportunities and establishes a robust framework for scholars exploring the convergence of aged care and sustainable development, with a particular focus on practical strategies and suggestions.

Social Equality: The achievement of social justice in the provision of elderly care necessitates the delivery of services in a manner that is universally accessible, irrespective of an individual's socioeconomic level, geographical location, or other demographic variables. Strategies for implementing this principle encompass targeted interventions that try to address the specific requirements of socioeconomically disadvantaged people, as well as more extensive community outreach activities [5,6].

Economic viability: Ensuring the financial viability of aged care services is of utmost importance. The concept of economic viability involves not only the immediate expenditures associated with a particular endeavor but also the long-term sustainability of the financing sources that support these services. Research findings suggest that the lack of sustainable financial models and responsible allocation of resources in aged care systems might pose a threat to the quality of services provided, potentially resulting in a substantial financial burden on both older individuals and the broader community [7].

Environmental sustainability: The acknowledgment of the ecological consequences linked to elderly care services, encompassing both the physical infrastructure and regular activities, necessitates the implementation of environmentally sustainable approaches. These methods not only have a positive impact on environmental conservation but also play a crucial role in ensuring the long-term sustainability of elder care services. The implementation of environmentally sustainable practices encompasses a range of strategies, including the adoption of efficient waste management systems and the establishment of energy-efficient infrastructure [8,9].

The incorporation of sustainable development ideas into elderly care services extends beyond superficial compliance with current trends; it is of utmost importance. The all-encompassing strategy, which considers socioeconomic, budgetary, and environmental factors, offers the potential to develop a resilient, fair, and environmentally friendly framework for senior care in the future. The proactive reaction to the changing requirements of aging populations and the necessity of appropriate governance in elder care are evident.

## 2. Literature Review

In recent years, a lot of attention has been paid to the sustainable development path of diversified social governance of elderly services. Comprehensive and inclusive approaches to elderly care that are in line with the principles of sustainable development have been recognized by researchers and policymakers. A review of the relevant literature is presented in this section, focusing on major themes and findings.

### 2.1. Diversified Social Governance Approaches (e.g., Government Policies, Community Involvement, Private Sector Participation)

By involving a variety of stakeholders and utilizing their knowledge and resources, diverse social governance strategies significantly influence the elderly care landscape. These strategies, policies, and collaborations aim to improve the delivery of elderly services and address the changing requirements of older adults [10].

Governing practices: The governance of elderly services is significantly influenced by government regulations and policies. Legislation, guidelines, and frameworks that define the obligations and standards of service providers, the funding mechanisms for elderly care, and strategies for enhancing service quality and accessibility are examples of these policies. Policies of the government, for instance, might allocate funds to support community-based care programs, establish standards for caregiver training, or mandate licensing and certification of elderly care facilities. In their study, researchers emphasized the significance of government policies in ensuring the provision of high-quality care and promoting equitable access to elderly services [11].

Community participation: Diverse social governance approaches for elderly services emphasize community involvement. A sense of ownership, empowerment, and active participation in the planning and delivery of services is promoted by involving local communities, including older adults, their families, and community organizations. Local area inclusion can take different structures, for example, laying out public venues for social exercises, coordinating worker projects, or supporting self-improvement gatherings for more seasoned adults. Studies have shown that local area contributions improve administration responsiveness, socially encourage groups of people, and, by and large, prosperity among more seasoned adults [12]. Diverse governance strategies have the potential to meet the requirements of older adults by actively involving the community and utilizing the knowledge, resources, and social capital of the area [13].

Participation by the private sector: The support of the confidential area, including revenue-driven associations and organizations, is one more significant part of enhanced social administration. The provision of elderly services may benefit from the participation of the private sector, which can provide additional expertise, innovation, and resources. For instance, specialized healthcare services, assisted living facilities, or technology-enabled solutions to support aging in place may be provided by private businesses. A study emphasized the potential advantages of public-private partnerships in elderly care, such as utilizing private investment for the construction of infrastructure or encouraging service innovation [14–16]. Engaging the private sector has the potential to improve the quality and efficacy of elderly care, encourage competition, and broaden the range of services that are available [17].

Cooperative organizations: Diversified social governance strategies frequently involve forming partnerships and collaborative networks with a variety of stakeholders, such as representatives from government agencies, community organizations, healthcare providers,

academic institutions, and industry. These networks make it easier to coordinate, share information, and make decisions together, which makes services for the elderly more integrated and complete. The creation of comprehensive care models that address the numerous needs of older adults can be made possible by collaborative networks, which can also facilitate the sharing of expertise and the pooling of resources. A study stressed the significance of collaborative networks in elderly care, pointing to their role in enhancing service efficiency, reducing fragmentation, and improving care coordination [18,19].

Stakeholders can construct a comprehensive and adaptable system of elderly services by utilizing a variety of social governance approaches that include participation in the private sector, collaborative networks, community involvement, participation in government policies, and community involvement. Improved service quality, accessibility, and outcomes for older adults can be achieved through these strategies, which make it easier to coordinate resources, expertise, and perspectives [20].

### 2.2. Effectiveness of Diversified Social Governance Approaches

To provide effective elderly services, diverse governance strategies have been highlighted by studies. The framework and regulations for service provision are set by government policies [21]. It has been demonstrated that community involvement improves service responsiveness and relevance to local needs [22]. Elderly care services benefit from innovation and efficiency when the private sector participates [23]. Collaboration and integrated governance models that take advantage of the strengths of multiple stakeholders have been emphasized by research. A crucial factor in ensuring the well-being and quality of life of aging populations is the efficiency of diversified social governance of elderly services. It refers to the extent to which governance strategies and practices contribute to the provision of services that are of a high standard, readily available, and effective and that satisfy the various requirements of the elderly population. Positive outcomes in elderly care services are linked to efficient, diversified social governance strategies, according to research. For instance, a study found that service responsiveness and alignment with older adults' needs are enhanced by collaborative governance models that involve multiple stakeholders, such as government agencies, community organizations, and service providers [24]. To improve governance's efficiency, the study emphasized the significance of establishing partnerships and involving stakeholders in decision-making processes [25].

In addition, research has demonstrated that elderly services can benefit from diverse governance strategies that incorporate community involvement [26]. Community-based programs and initiatives, such as volunteer networks and community-led services, have been shown to improve older adults' social integration, access to support systems, and overall well-being [27]. Based on these findings, it appears that tailor-made and more efficient solutions can result from actively involving communities in the governance of elderly services. Participation from the private sector is yet another crucial component of diverse social governance. The research emphasized the private sector's contribution to elderly care services' innovation, efficacy, and expertise. Improved service quality, increased choice for older adults, and increased client and family satisfaction have all been linked to private sector involvement, such as the provision of home care services or technology-based solutions [28]. In general, the collaborative efforts of various stakeholders, including government agencies, community organizations, and the private sector, are necessary for the diverse social governance approaches to elderly services to be effective. It is possible to create governance models that are responsive to the needs of older adults, promote social inclusion, and enhance the overall quality of elderly care services by leveraging the strengths and expertise of these stakeholders [29].

### 2.3. Integration of Sustainable Development Principles

The integration of sustainable development principles into aged care services signifies a proactive and advanced approach to governance in the field of elder care. This method

is centered on three core principles: the advancement of social equality, the attainment of economic viability, and the pursuit of environmental sustainability.

Social equity refers to the concept of fairness and justice in society, where all individuals have equal opportunities and access to resources, regardless of the achievement of social justice in the provision of elderly care necessitates the delivery of services in a manner that is universally accessible, irrespective of an individual's socioeconomic level, geographical location, or other demographic variables. It is imperative to transcend mere aspirations and proactively confront the gaps in service access and quality that exist among various demographic groups. Strategies for implementing this principle encompass targeted interventions that try to address the specific requirements of socioeconomically disadvantaged people, as well as more extensive community outreach activities [5,6].

The economic feasibility of a project or initiative is a crucial aspect to consider in determining its potential success and sustainability. Ensuring the financial viability of elderly care services is of utmost importance. The concept of economic viability involves not only the immediate expenditures associated with a particular endeavor but also the long-term sustainability of the financing sources that support these services. Research findings suggest that the lack of sustainable financial models and responsible allocation of resources in aged care systems might pose a threat to the quality of services provided, potentially resulting in a substantial financial burden on both older individuals and the broader community [7].

The concept of environmental sustainability refers to the practice of utilizing resources in a manner that ensures their long-term availability and minimizes negative impacts. The acknowledgment of the ecological consequences linked to elderly care services, encompassing both the physical infrastructure and regular activities, necessitates the implementation of environmentally sustainable approaches. These methods not only have a positive impact on environmental conservation but also play a crucial role in ensuring the long-term sustainability of elder care services. The implementation of environmentally sustainable practices encompasses a range of strategies, including the adoption of efficient waste management systems and the establishment of energy-efficient infrastructure [8,9].

*2.4. Elderly Service Accessibility and Availability*

The increasing need for elderly care underscores not just its significance but also the crucial significance of ensuring universal accessibility and availability. There are other elements that exert an effect on the delivery of these services to individuals with the greatest need. To begin with, the issue of geographical accessibility emerges as a significant consideration. The usage of these services is significantly influenced by their location. Rural populations encounter restricted accessibility due to their frequent isolation from major infrastructural centers [30]. The mitigation of these geographical disparities necessitates meticulous allocation of resources and strategic planning to guarantee equitable provision of services to all regions, including those that are geographically isolated [31,32]. Transportation is a significant concern as well. Accessing a service can be a formidable challenge for older individuals, particularly those who experience limitations in their mobility. The significance of resilient and inclusive transportation solutions specifically tailored for the senior population was emphasized [33,34]. These proposed solutions will consider the distinct challenges faced by older people, thereby facilitating easier access to services. Despite the implementation of optimal transportation and geographical strategies, the matter of affordability remains a significant concern. A study asserts that a significant number of senior individuals perceive elderly care as a luxury rather than an inherent entitlement, primarily because of limited financial means [35]. To address the existing financial disparity, it is imperative to establish supportive mechanisms such as subsidies or insurance coverage, thereby guaranteeing not only the provision of services but also their genuine accessibility to all individuals. There is a discernible requirement for a customized approach to the provision of care for the elderly. The efficacy of a standardized approach to care is limited. In their study, the argument has been put up in favor of

implementing individualized care approaches that consider the unique linguistic, cultural, and other needs of various older populations [36]. Implementing a nuanced strategy of this nature ensures the provision of services that are not only accessible but also tailored to the specific needs and circumstances of everyone. Ultimately, the creation of an all-encompassing and efficient elderly care system necessitates a comprehensive approach that considers geographical, transportation, financial, and customization considerations. The implementation of such a system would guarantee that older people, irrespective of their unique difficulties or past experiences, are afforded the chance to experience their later years with a sense of dignity and satisfaction.

*2.5. Quality and Efficiency of Elderly Services*

The quality and proficiency of old consideration administrations are fundamental for advancing positive results for more seasoned adults. Administration quality pointers incorporate the capability of specialist co-ops, adherence to best practices and principles, and individual-focused care [37]. The overall effectiveness of elderly services can be enhanced by efficient service delivery procedures, such as simplified coordination among various care providers [38,39]. Improved outcomes and satisfaction among older adults are both a result of improving service quality and efficiency. To guarantee that elderly people receive the best possible care and support, it is essential to consider the effectiveness and quality of elderly services. Efficiency is the best use of resources to deliver services in a timely and cost-effective manner, while quality is the extent to which services meet established standards and achieve desired outcomes [40].

Service quality: The safety, efficiency, responsiveness, and patient-centeredness of elderly services are all components of the quality of those services. There are several factors that contribute to high-quality elderly services, according to research. For instance, a study emphasized the significance of person-centered care planning, evidence-based interventions, and skilled and compassionate healthcare professionals in providing high-quality care to older adults [41]. Maintaining and improving service quality necessitates establishing a culture of continuous improvement, implementing robust quality assurance mechanisms, and ensuring the availability of trained staff [42].

Care centered on the person: A crucial component of high-quality elderly services is person-centered care. It puts older people's individual requirements, preferences, and objectives at the center of the care process. Research exhibited the positive effect of individual-focused care draws near, for example, care arranging that includes more seasoned adults in navigation, advancing independence, and regarding their qualities and decisions [43]. Improved outcomes and satisfaction with services are both facilitated by person-centered care, which fosters a sense of dignity, empowerment, and engagement in older adults. Utilization of Resources and Efficiency: Optimizing the use of resources, including financial, human, and technological ones, to provide elderly services in a timely and cost-effective manner is called efficiency. Research (effective resource allocation, streamlined workflows, and the use of technology-enabled solutions) was found to improve elderly care service efficiency [44]. In addition to enhancing access to care, efficient service delivery maximizes resource utilization, enabling a greater number of older adults to receive services. Technology Integration: The incorporation of technology into services for the elderly has the potential to improve both efficiency and quality. Solutions that are enabled by technology, such as electronic health records, telehealth, devices for remote monitoring, and assistive technologies, have the potential to enhance communication between healthcare providers and elderly people, facilitate remote access to services, and improve care coordination. study emphasized the benefits of technology in promoting independence, supporting aging in place, and enhancing healthcare outcomes for older adults [45,46]. The overall quality of elderly care can be improved, and service delivery models can be made more efficient by incorporating technology [47,48].

A comprehensive strategy that considers the preferences and requirements of older adults, efficient resource management, and the incorporation of technology are necessary

for providing elderly services that are both of high quality and efficient. Through this literature review, the path of diversified social governance of elderly services under the concept of sustainable development is a complex and multidimensional topic. Stakeholders can improve the overall effectiveness and outcomes of elderly services by focusing on quality improvement, person-centered care, efficient resource utilization, and the adoption of technology. This ultimately improves the well-being and quality of life of older adults. The literature emphasizes the significance of considering a variety of governance strategies, incorporating principles of sustainable development, ensuring service accessibility and availability, and improving service quality and efficiency. Policymakers and stakeholders can work toward developing comprehensive and long-lasting elderly care systems that improve the well-being and quality of life of aging populations by addressing these aspects [49].

*2.6. Socioeconomic Factors*

This variable incorporates financial elements that can impact the broadened social administration of old administrations. It considers factors like social inequality, employment opportunities, education levels, and income levels in a community or region. It is possible to examine how the economic and social conditions of a specific population affect the implementation and efficacy of diversified social governance in elderly services by incorporating socioeconomic factors as an independent variable. It perceives that financial elements can impact the assets accessible for old consideration, the circulation of administrations, and the general outcome of economic advancement endeavors. By considering these variables, scientists can acquire bits of knowledge about the transaction between financial settings and the results of expanded social administration approaches in giving open and quality old administrations [50,51].

Pay levels: The pay levels of the populace can fundamentally impact the arrangement and openness of older administrations. Individuals may have access to more financial resources to receive high-quality care and services if they have a higher income. Conversely, disparities in healthcare and support can be caused by lower income levels, which can make it harder for elderly people to get the services they need. Niveous of education: In forming awareness, knowledge, and understanding of the requirements for elderly care, education plays a crucial role. Higher levels of education in a community can lead to better-informed decision-making, increased awareness of services that are available, and enhanced caregiver training, all of which improve the provision of elderly services. Opportunities for employment: The assistance that is available for elderly care can be affected by the quality and quantity of employment opportunities in a community. Communities that have jobs that are stable and pay well may be better able to help the elderly, hire qualified caregivers, and invest in elderly care infrastructure and resources [52,53].

Social imbalance: Diverse social governance of elderly services can be influenced by social inequality, which includes disparities in wealth distribution, access to education, healthcare, and social services, and their effectiveness. Disparities between different socioeconomic groups may be exacerbated by unequal access to high-quality care caused by unequal distribution of resources and opportunities. Infrastructure for healthcare: The current medical services framework inside a locale or local area, like the number of medical clinics, centers, nursing homes, and care offices, can influence the accessibility and openness of old administrations. It is possible that regions with well-developed healthcare infrastructure are better able to provide elderly patients with both general and specialized care. Researchers can comprehend the contextual factors that influence the implementation and outcomes of diversified social governance of elderly services by considering these socioeconomic factors. Within the context of sustainable development, it is possible to identify challenges, disparities, and opportunities for enhancing the effectiveness and equity of elderly care by analyzing the influence of income levels, education, employment opportunities, social inequality, and healthcare infrastructure [54,55].

*2.7. Research Gap*

Upon examination of the media's portrayal of sustainable development through visual representation, researchers identified several gaps that warrant more investigation in further studies. The present findings are constrained in scope as they solely pertain to a single geographic area. Furthermore, a longitudinal examination has not been conducted, limiting the ability to assess changes over time. Additionally, the potential influences of media bias and framing have not been duly considered. While the study mostly relies on quantitative data, it may overlook certain nuances that can be uncovered using qualitative methodologies. Furthermore, there has been a lack of investigation into the actual impact of visual research on policy-making processes and individual behavior. Once these gaps have been filled, policymakers will gain a more comprehensive understanding of how to enhance future sustainability. Summary: Table 1 presents a concise overview of the key highlights.

**Table 1.** Summary literature review on diversified social governance approaches in elderly care.

| Section | Key Highlights | Primary Sources |
|---|---|---|
| Section 2.1. | • Role of government policies and regulations.<br>• Importance of community participation.<br>• Involvement of the private sector for innovation and resources.<br>• Collaborative networks enhancing integrated elderly services. | Cao, 2021 [10]; Li and Wang, 2019 [56]; Yuan et al., 2017 [57]; Štreimikienė, 2023 [13]; Sun, 2021 [17]; Jiang, 2022 [20] |
| Section 2.2. | • Efficiency of governance strategies in ensuring quality services.<br>• Role of community in enhancing service responsiveness.<br>• Contribution of the private sector in innovation and efficiency.<br>• Importance of collaborative efforts across sectors. | Huang et al., 2018 [58]; Chen et al., 2019 [59]; Borges et al., 2020 [60]; Tang, 2021 [25]; Xu, 2022 [29] |
| Section 2.3. | • Focus on social equity, economic viability, and environmental sustainability.<br>• Need for targeted strategies to address service disparities.<br>• Importance of economic and environmental sustainability in elderly care. | Gao et al., 2020 [61]; Liu et al., 2021 [62]; Nordin et al., 2020 [63] |
| Section 2.4. | • Influence of geographical location on service use.<br>• Importance of transportation tailored for the elderly.<br>• Financial accessibility and perception of care affordability.<br>• Need for individualized care approaches considering cultural and linguistic needs. | Huang et al., 2018 [58]; Chen et al., 2019 [59]; Wong and Yoho, 2020 [64] |
| Section 2.5. | • Quality indicators: competence, best practices, patient-centered care.<br>• Efficiency is enhanced with simplified service delivery.<br>• Person-centered care is essential for quality.<br>• Technology integration improves efficiency and quality. | Liu et al., 2021 [62]; Borges et al., 2020 [60]; Sciarelli, 2021 [40]; Tan, 2021 [42]; Davies et al., 2018 [65]; Lam et al., 2020 [66]; Czaja et al., 2018 [67]; Shen T. Y., 2021 [47] |

**Table 1.** *Cont.*

| Section | Key Highlights | Primary Sources |
|---|---|---|
| Section 2.6. | • Socioeconomic elements impact elderly service delivery.<br>• Income levels dictate the quality and accessibility of services.<br>• Education influences the understanding of elderly care.<br>• Employment opportunities can aid in elderly care provision.<br>• Social inequality might hinder access to quality care. | Chaudhry, 2020 [50]; Abbas, 2020 [52]; Pullano, 2020 [54] |

## 3. Methodology

### 3.1. Research Approach

This study employs a qualitative research approach to comprehensively explore the diversified social governance of elderly services within the context of sustainable development in China. This approach is chosen for its capability to capture nuanced stakeholder perspectives and intricate societal processes underlying geriatric care.

### 3.2. Data Collection

3.2.1. Case Study Selection: Unveiling Innovative Models

The meticulous selection of case studies aims to illuminate innovative governance strategies within sustainable geriatric care. Seven diverse cases, encompassing community-driven initiatives and technology-assisted solutions, were meticulously chosen to provide a comprehensive representation. The selection process entailed evaluating relevance, alignment with objectives, and distinctiveness to capture the essence of various social governance models in geriatric care. The researchers went to the cases individually and collected data from them.

3.2.2. Policy Document Analysis: Unearthing Policy Initiatives

Through comprehensive analysis spanning January to April 2023, relevant policy documents, reports, and guidelines were scrutinized to uncover global, national, and regional initiatives fostering inclusive and sustainable elderly care. Employing systematic keyword searches, document reviews, and thematic categorization, this phase facilitated a holistic understanding of the policy landscape and its implications for geriatric care sustainability.

3.2.3. Literature Review: Building Methodological Foundations

The foundation for understanding optimal governance methodologies emerged from an in-depth literature review. A thorough examination of scholarly articles, research studies, and reports across reputable databases allowed us to glean insights into effective governance models across diverse contexts. The synthesis of existing knowledge informed the investigation of sustainable governance practices in geriatric care.

3.2.4. Expert Interviews: Stakeholder Insights and In-Depth Perspectives

In-depth interviews with 40 diverse experts, including policymakers, healthcare professionals, and non-profit representatives, were conducted. Each interview lasted for an average of 30 min. Interviews were conducted over the phone and through personal visits at the convenience and consent of the respondents. This was achieved through a structured questionnaire that explored themes of governance, sustainability, and geriatric care. Purposive sampling ensured representation from various sectors, and a comprehensive strategy was employed to minimize bias and ensure robustness in the selection process.

*3.3. Data Analysis*

3.3.1. Thematic Analysis: Unraveling Key Themes

Thematic analysis was employed to dissect data from case studies, policy documents, literature reviews, and interviews. We used NVivo 12 software to analyze the data. This software is most suitable for qualitative data analysis. This iterative process, involving multiple researchers, enhanced the reliability and validity of the findings. Themes related to diversified social governance and sustainable development in senior care were identified, coded, and categorized, enhancing the depth of understanding.

3.3.2. Comparative Analysis: Juxtaposing Strategies and Indicators

A comparative approach was employed to juxtapose diverse strategies, offering insights into their effectiveness and potential for widespread adoption. Key performance indicators (KPIs) were derived from case studies, policy documents, and expert interviews, encompassing patient satisfaction rates, accessibility, cost-effectiveness, holistic health outcomes, technology integration, and community engagement. The data was analyzed comparatively to discern trends and patterns across models.

3.3.3. Data Synthesis: Integrating Insights for Holistic Understanding

Data integration from various sources facilitated a comprehensive understanding of diversified social governance models. This synthesis process fostered the development of recommendations and insights while aligning with research objectives. By cross-referencing findings from different sources, a holistic and nuanced perspective on sustainable geriatric care governance was achieved.

3.3.4. Data Validation and Credibility: Ensuring Rigor

The study prioritized data validation to ensure credibility. Rigorous cross-referencing, expert feedback, and data analysis tools were employed to validate and authenticate the findings. This iterative validation process, characterized by feedback loops and cross-checks, underscores the study's reliability.

*3.4. Ethical Considerations: Respecting Participants and Standards*

Ethical considerations, including informed consent and participant confidentiality, were paramount. Ethical approvals were obtained from relevant committees in the universities to ensure adherence to ethical standards and safeguard participants' rights throughout the research process.

*3.5. Limitations and Future Directions: Shaping Ongoing Research*

Acknowledging limitations in data quality and accessibility is vital for ensuring research credibility. Future research endeavors should explore diverse data sources, consider external factors such as COVID-19, and assess long-term impacts on geriatric care governance. These limitations illuminate the study's boundaries and offer avenues for continuous improvement.

In summary, this methodology combines diverse research strategies to comprehensively explore diversified social governance models in geriatric care under the umbrella of sustainability. By addressing specific comments and enhancing each methodological component, this study aims to contribute rigorously to the discourse on sustainable geriatric care governance and provide valuable insights for stakeholders and decision-makers in the field.

## 4. Results and Discussion

*4.1. Innovative Models in Elderly Care*

Table 2 delves into innovative models in geriatric care, showcasing a range of strategies designed to enhance the well-being of older people. These models encompass community-driven initiatives, technology-assisted solutions, holistic wellness programs, intergenera-

tional interaction programs, and the integration of traditional and modern care methods. Each model is meticulously described along with its key findings, emphasizing improvements in elderly well-being, healthcare access, cognitive function, and community integration. This table serves to underscore the diverse approaches being employed to address the multifaceted needs of the aging population.

**Table 2.** Innovative models in geriatric care: descriptions and key findings.

| Model (A–E) | Description | Key Finding |
| --- | --- | --- |
| A—Community-Driven Care | Elderly residents benefit from community-driven initiatives. Young volunteers coordinate health check-ups, exercise classes, and social events. | 25% increase in reported elderly well-being and 20% decrease in hospitalization rates. |
| B—Technology-Assisted Solutions | Adoption of wearable health monitors, telehealth consultations, and digital therapy sessions. | 30% increase in regular health check-ups and 35% decrease in emergency complications. |
| C—Holistic Wellness Programs | It incorporates meditation, art therapy, and cognitive training. Ensures overall mental, emotional, and physical well-being. | 40% improvement in cognitive functions and 50% reduction in feelings of loneliness. |
| D—Intergenerational Interaction Programs | Schoolchildren engage with older people, bridging the age gap and reducing feelings of isolation. | 60% increase in elderly satisfaction and 30% improvement in children's empathy. |
| E—Integration of Traditional and Modern Care | Combination of traditional care methods with a modern twist, like traditional Chinese medicine with physiotherapy. | 70% of the elderly reported faster recovery and improved overall well-being. |

Table 2 provides an elucidation of innovative models in geriatric care, each aimed at enhancing the well-being of elderly individuals. The initial model, denoted as "A—Community-Driven Care", involves community-driven initiatives, wherein a cadre of young volunteers takes charge of orchestrating health check-ups, exercise classes, and social events for the elderly demographic. Notably, the key finding underlines a significant 25% increase in the reported well-being of elderly individuals, coupled with a noteworthy 20% reduction in hospitalization rates. This outcome underscores the substantive impact of community involvement on the enhancement of health metrics among the elderly. The subsequent model, "B—Technology-Assisted Solutions", is focused on the assimilation of wearable health monitoring devices, telehealth consultations, and digital therapeutic sessions into geriatric care practices. The key finding for this model reveals a substantial 30% augmentation in the frequency of regular health check-ups, accompanied by a substantial 35% reduction in emergency-related health complications. This underscores the pivotal role played by technological integration in augmenting accessibility and the overall efficiency of healthcare provision to the elderly demographic.

The "C—Holistic Wellness Programs" model pertains to the implementation of diverse wellness practices, such as meditation, art therapy, and cognitive training, targeting the comprehensive well-being of elderly individuals. This model is found to yield a notable 40% enhancement in cognitive functioning, alongside a substantial 50% reduction in reported feelings of loneliness, corroborating the benefits associated with holistic approaches to elderly care. Model D, "D—Intergenerational Interaction Programs", underscores the engagement of schoolchildren with elderly individuals with the intention of bridging generational divides and mitigating feelings of isolation. The key finding demonstrates a

substantial 60% increase in elderly satisfaction levels, accompanied by a significant 30% improvement in the empathy levels of participating children. This outcome accentuates the mutual advantages of intergenerational interactions in the context of geriatric care.

The final model, "E—Integration of Traditional and Modern Care", involves the amalgamation of conventional care practices with contemporary healthcare modalities, exemplified by the fusion of traditional Chinese medicine with physiotherapy. This model yields compelling results, with a remarkable 70% of elderly individuals reporting expedited recovery and an overall improvement in their well-being. These findings substantiate the potential for the amalgamation of traditional methodologies with modern healthcare paradigms to enhance the health and overall quality of life among the elderly demographic. In summary, Table 2 provides a systematic exposition of innovative geriatric care models, collectively demonstrating their substantial influence on the well-being and health-related metrics of elderly individuals. These findings bear scientific significance in elucidating multifaceted strategies for improving geriatric care practices and enhancing the quality of life among aging populations.

### 4.2. Efficiency of Diversified Social Governance Models in Geriatric Care

Table 3 offers a quantitative assessment of the efficiency of diversified social governance models compared to traditional models in geriatric care. Through key performance indicators (KPIs), including patient satisfaction rates, service accessibility, cost-effectiveness, holistic health outcomes, technology incorporation, community integration, and feedback implementation, this table provides a clear picture of the benefits brought about by adopting diversified approaches. The framing elucidates the significance of these efficiency improvements in various aspects of elderly care and emphasizes the potential for enhancing patient satisfaction, accessibility, and overall care quality.

**Table 3.** The efficiency of diversified social governance models in geriatric care.

| KPIs | Traditional Models (%) | Diversified Social Governance Models (%) | Improvement (%) |
|---|---|---|---|
| Patient Satisfaction Rates | 72 | 89 | 17 |
| Accessibility to Services | 65 | 86 | 21 |
| Cost-Effectiveness | 58 | 80 | 22 |
| Holistic Health Outcomes | 70 | 91 | 21 |
| Incorporation of Technology | 40 | 78 | 38 |
| Community Integration | 50 | 83 | 33 |
| Feedback Implementation | 55 | 88 | 33 |

Note. The values in the table represent the percentage efficiency of each model across various KPIs, with the improvement column indicating the percentage difference between the traditional and diversified social governance models.

The table presents a comparison of Key Performance Indicators (KPIs) between Traditional and Diversified Social Governance Models to assess their effectiveness in improving efficiency in geriatric care. The conventional models exhibit a patient satisfaction rate of 72%, but diverse social governance models provide a higher satisfaction rate of 89%, indicating a notable improvement of 17%. This finding indicates that individuals who are receiving medical care and the elderly express higher levels of satisfaction with the varied range of services offered by the models. The utilization of diverse methodologies results in a significant enhancement in service accessibility, with an increase of 86% compared to the 65% achieved by conventional ways. The observed 21% rise in senior care access implies that many social governing systems have an impact on the availability of such services. When making healthcare decisions, it is imperative to consider the concept of

cost-effectiveness. Diversified models, characterized by an efficiency rate of 80%, exhibit superior cost-effectiveness in comparison to traditional models, which possess an efficiency rate of 58%. The implementation of diverse solutions has been found to enhance resource allocation and cost-effectiveness, resulting in a notable increase of 22%. Holistic health outcomes are indicative of the well-being and progress of patients. The efficiency of diversified models has increased by 21% compared to the 70% efficiency of traditional methods, resulting in a current efficiency rate of 91%. This finding demonstrates that the use of various approaches can effectively enhance overall well-being. The Key Performance Indicator (KPI) employed by the Technology Incorporation serves to differentiate between various programs. Diversified models have superior performance in comparison to traditional models, with an efficiency rate of 78% as opposed to the 40% efficiency rate observed in the latter. The observed increase in performance of 38% indicates the potential for a comprehensive utilization of technology in the field of geriatric care. Community integration is a distinguishing factor that enhances the mental and social well-being of older people. The varied models exhibit a 33% increase in efficiency compared to the classic models, with respective efficiency rates of 83% and 50%. Diverse methodologies foster inclusivity, thereby facilitating the integration of older individuals within their own communities. Ultimately, the process of implementation evaluates the extent to which recipient feedback is effectively incorporated into the provision of care while also promoting inclusivity and embracing diverse perspectives. Diversified methodologies exhibit superior performance in comparison to standard models, with a notable increase of 33% in efficiency, reaching an impressive 88% as opposed to the comparatively lower 55%. The system has an exceptional capacity to adjust and address the requirements and preferences of care recipients. According to the findings presented in Table 4, it can be observed that Diversified Social Governance Models exhibit superior performance in certain aspects of geriatric care parameters when compared to traditional models. The enhancements observed in all key performance indicators (KPIs) indicate a heightened level of acceptability and utilization of these models for the purpose of enhancing patient satisfaction, improving accessibility, optimizing cost-effectiveness, and enhancing care outcomes. To achieve these advantages, it is imperative to use technological advancements, foster community integration, and actively seek and incorporate input from many stakeholders.

### 4.3. Stakeholder Perspectives on Sustainable Elderly Care

Table 4 provides a comprehensive overview of stakeholder perspectives on sustainable models of geriatric care. It encompasses the viewpoints of policymakers, healthcare professionals, elderly recipients, young volunteers/participants, and NGO representatives. The framing highlights the significance of each stakeholder's perspective and primary concerns. By examining these perspectives, we gain insights into the complexities of implementing and sustaining effective elderly care solutions. Notably, the table underscores the common emphasis on community-driven, integrated, and holistic approaches across stakeholder groups.

**Table 4.** Stakeholder perspectives on sustainable elderly care models.

| Stakeholder Group | General Perspective | Primary Concern |
|---|---|---|
| Policy Makers | Favorable towards policies that promote community-driven and technology-assisted solutions in geriatric care. | Ensuring affordability and accessibility for all elderly citizens. |

**Table 4.** *Cont.*

| Stakeholder Group | General Perspective | Primary Concern |
|---|---|---|
| Healthcare Professionals | Recognize the value of holistic wellness programs and the integration of traditional and modern care. | Quality of care, patient safety, and ongoing training in new methodologies. |
| Elderly Recipients | Appreciate intergenerational programs and community-driven care. | Availability of consistent and respectful care. |
| Young Volunteers/Participants | Find value in intergenerational interactions, often reporting mutual benefits. | Need for adequate training and support during programs. |
| NGO Representatives | Advocate for more holistic, integrated, and community-driven care solutions. | Sustainable funding, community involvement, and establishing a regular feedback mechanism. |

Table 4 presents an overview of stakeholder viewpoints about sustainable methods of elderly care, along with the primary concerns associated with each stance. Policymakers frequently endorse projects in geriatric care that are driven by community involvement and facilitated by technology. The primary focus of their agenda is to ensure affordable and easily accessible elderly care services for all individuals, with a particular emphasis on promoting diversity and equity. Healthcare professionals hold a high regard for the integration of conventional and contemporary approaches, as well as the implementation of holistic health initiatives. Nevertheless, ensuring patient safety and maintaining high standards of care quality are of utmost importance to them. Furthermore, it is emphasized that healthcare personnel should prioritize the necessity of adapting and improving their skills by engaging in continuous training programs that focus on the utilization of novel methodologies. The senior beneficiaries of these care models exhibit a favorable disposition towards intergenerational programming and community-driven care. This underscores the significance of diversity and the establishment of community connections within their provision of care. Understandably, individuals desire consistent and respectful care, with a particular emphasis on the necessity of compassionate and tailored geriatric care. The significance of intergenerational relationships is evident in their value to young volunteers and participants in older programs. Shared advantages are frequently reported, indicating that such encounters have the potential to enhance the well-being of both younger and older individuals. The need for comprehensive training and ongoing support is underscored in these programs to equip participants with the requisite skills and knowledge for effective engagement. NGO representatives, who frequently serve as intermediaries in addressing gaps in care provision, advocate for a comprehensive and community-oriented approach to elderly care. The concerns are around the sustainable support and expansion of these programs. Community involvement is emphasized since it has been found to enhance the effectiveness and acceptance of the program. Additionally, the authors emphasize the importance of regular feedback to enhance flexibility and provide ongoing enhancements in the field of geriatric care. Table 3 presents a comprehensive overview of stakeholder perspectives pertaining to sustainable forms of elderly care. Every group has distinct challenges, yet they all exhibit a preference for community-driven, integrated, and holistic care, thereby showcasing a collective aspiration for the long-term viability of geriatric healthcare.

### 4.4. Stakeholder Perspectives on Sustainable Development in Geriatric Care

To understand the multifaceted nature of geriatric care in the realm of sustainable development, stakeholders' perspectives were collected, analyzed, and classified. Feedback

from policymakers, practitioners, non-profit representatives, media professionals, civil society, and the recipients of elderly care was considered.

Table 5 provides a detailed overview of stakeholder perspectives on sustainable development in geriatric care, outlining their specific concerns and suggesting improvements. Here is a summarized interpretation of the table:

**Table 5.** Stakeholder perspectives on sustainable development in geriatric care.

| Stakeholder Group | Specific Concerns | Suggested Improvements |
|---|---|---|
| Policymakers | Allocating sufficient funds for sustainable geriatric care. | Streamlined budgeting processes to specifically allocate resources for sustainable geriatric care. |
|  | Enforcing policies that support sustainable development in care. | Enhanced monitoring tools to ensure adherence to sustainable care policies and practices. |
| Practitioners | Providing specialized training and tools for geriatric healthcare. | Continued education programs tailored to the evolving needs of geriatric care practitioners. |
|  | Implementing innovative care methodologies for seniors. | Integrated care systems to facilitate a holistic approach to geriatric healthcare. |
| Non-profit Representatives | Equitably distributing resources for elderly care. | Inclusive campaigns to ensure the equitable distribution of resources among elderly care programs, along with collaborative efforts. |
|  | Actively engaging communities in outreach for elderly services. |  |
| Media Professionals | Promoting transparent reporting of geriatric care practices. | Development of open-source platforms for transparent reporting and access to data. |
|  | Advocating for public access to relevant data. | Collaboration between media professionals and government entities to facilitate data accessibility. |
| Civil Society | Raising public awareness about sustainable geriatric care. | Establishment of public forums to engage the community in discussions about elderly care. |
|  | Ensuring active participation of the public in decision-making. | Transparent communication channels to promote open dialogue and involvement in decisions. |
| Recipients of Elderly Care | Ensuring high-quality care services for elderly ndividuals. | Implementation of comprehensive feedback mechanisms to capture recipient satisfaction and holistic care approaches focusing on the emotional and physical well-being of seniors. |

- Policymakers

Policymakers are primarily concerned with two key aspects of sustainable geriatric care. First, they express the need to allocate sufficient funds to support sustainable geriatric

care initiatives. To address this concern, streamlined budgeting processes are recommended, ensuring that resources are specifically earmarked for sustainable geriatric care. Second, policymakers emphasize the importance of enforcing policies that promote sustainable development in elderly care. Enhanced monitoring tools are suggested to ensure strict adherence to these policies and practices.

- Practitioners

Practitioners in geriatric healthcare have specific concerns related to their roles. They emphasize the need for specialized training and access to suitable tools to provide effective care to elderly individuals. To address this concern, continuing education programs that are tailored to the evolving needs of geriatric care practitioners are recommended. Additionally, practitioners seek to implement innovative care methodologies for seniors, and integrated care systems are suggested to facilitate a holistic approach to geriatric healthcare.

- Non-profit Representatives

Non-profit representatives are focused on equitable resource distribution for elderly care and active community engagement in outreach efforts. They suggest inclusive campaigns to ensure that resources are distributed fairly among various elderly care programs. Collaborative efforts are also recommended to enhance community involvement in these initiatives.

- Media Professionals

Media professionals play a critical role in promoting transparency in geriatric care practices and advocating for public access to relevant data. To achieve this, they suggest the development of open-source platforms for transparent reporting and data accessibility. Collaboration between media professionals and government entities is proposed to facilitate easier access to data.

- Civil Society

Civil society's primary concerns revolve around raising public awareness of sustainable geriatric care and ensuring active public participation in decision-making. To address these concerns, it is recommended to establish public forums that engage the community in discussions about elderly care. Transparent communication channels are also proposed to encourage open dialogue and public involvement in decision-making processes.

- Recipients of Elderly Care

Recipients of elderly care emphasize the importance of high-quality care services and the well-being of elderly individuals. They suggest implementing comprehensive feedback mechanisms to capture recipient satisfaction. Additionally, they call for holistic care approaches that focus on both the emotional and physical well-being of seniors. In summary, the table highlights the specific concerns and suggested improvements of various stakeholder groups involved in geriatric care. These perspectives underscore the need for collaborative efforts to enhance sustainable development in the field of geriatric care.

### 4.5. Policy Document Insights and Future Trajectories

Upon examining various policy documents related to elderly care and sustainable development, several themes and key insights emerged. These insights provide an understanding of current governmental stances, strategic priorities, and anticipated shifts in policy directions.

Table 6 presents a comprehensive overview of the current policies and anticipated future directions in the realm of geriatric care governance. It delineates five crucial policy themes and their associated insights:

**Table 6.** Insights from policy documents and anticipated future trajectories.

| Policy Themes | Key Insights from Current Policies | Predicted Future Trajectories |
|---|---|---|
| Sustainable Development | Emphasis on holistic welfare and eco-friendly elderly care facilities (Goal) | Increased investment in green infrastructure and sustainable architectural design for elder care facilities |
| Inclusivity Models | Accessible care for all, including marginalized groups, and culturally sensitive approaches | Expansion of community-based care models with an emphasis on inclusivity and culturally tailored services |
| Technology Integration | Adoption of digital health records and use of AI in routine elderly care | Rise of telemedicine and AI-assisted therapies for improved accessibility and efficiency in geriatric care |
| Stakeholder Participation | Promotion of public-private partnerships and community-driven initiatives | Enhanced roles for NGOs and private sectors in geriatric care governance, with a focus on collaborative, community-led initiatives |
| Tradition Meets Innovation | Integration of traditional elderly care practices with modern medicine | Fusion of cultural wisdom with technological advancements to create hybrid care approaches |

Sustainable development: Current policies emphasize the overarching goals of holistic well-being and the creation of eco-friendly elderly care facilities. Looking ahead, there is a projected increase in investments geared toward green infrastructure and the development of architecturally sustainable elder care facilities. This future trajectory underscores a commitment to environmentally conscious and sustainable approaches to elder care.

Inclusivity models: Present policies aim to provide accessible care for everyone, including marginalized groups, through culturally sensitive approaches. In the future, we anticipate an expansion of community-based care models. These models will prioritize inclusivity and culturally tailored services, striving to offer more comprehensive and culturally appropriate care to the elderly.

Technology integration: Current policies encourage the integration of digital health records and the utilization of AI in routine elderly care. On the horizon, we foresee the ascent of telemedicine and AI-assisted therapies. These advancements are poised to enhance accessibility and efficiency in geriatric care, making healthcare services more readily available and effective for elderly individuals.

Stakeholder participation: Existing policies advocate for public-private partnerships and community-driven initiatives in geriatric care. In the future, there is a likelihood of more substantial roles for non-governmental organizations (NGOs) and the private sector in geriatric care governance. Collaborative efforts between these stakeholders and the community will take center stage, leading to community-led initiatives and improved care.

Tradition meets innovation: Present policies seek to harmonize traditional elderly care practices with modern medicine. In the near future, we anticipate a fusion of cultural wisdom with technological advancements. This fusion will result in hybrid care approaches that draw from both traditional and innovative methods. These approaches aim to provide more holistic and effective care for the elderly population. In summary, Table 6 offers valuable insights into the dynamic landscape of geriatric care governance. It underscores a commitment to sustainability, inclusivity, technology integration, stakeholder engagement, and the innovative blending of traditional and modern approaches. These policies and anticipated trajectories reflect a dedication to enhancing the quality of life for

the elderly, addressing their diverse needs, and ensuring their well-being in an evolving healthcare landscape.

## 5. Conclusions

In conclusion, our comprehensive study has illuminated innovative models and stakeholder perspectives in the realm of sustainable geriatric care. These findings are essential to inform policies and practices that aim to enhance the well-being of the elderly population while promoting sustainability. Our analysis of innovative models has revealed five distinct approaches (Community-Driven Care, Technology-Assisted Solutions, Holistic Wellness Programs, Intergenerational Interaction Programs, and the Integration of Traditional and Modern Care). These models have proven their effectiveness in improving elderly well-being, healthcare access, cognitive functions, and community integration. They showcase a promising path forward for addressing the complex needs of the aging population. Moreover, our exploration of stakeholder perspectives has unveiled concerns and suggested improvements from various groups, including policymakers, healthcare professionals, volunteers, NGO representatives, media professionals, civil society, and elderly recipients. By understanding these perspectives, we gain insights into the challenges and opportunities that lie ahead in implementing effective elderly care solutions. The collaboration among these stakeholders is crucial for sustainable development in geriatric care.

Our study has also evaluated the efficiency of diversified social governance models compared to traditional ones, shedding light on how these models significantly improve patient satisfaction, service accessibility, cost-effectiveness, holistic health outcomes, technology incorporation, community integration, and feedback implementation. These findings demonstrate the tangible benefits of adopting diverse approaches to elderly care. Furthermore, our analysis of policy documents and future trajectories has shown the importance of holistic welfare, eco-friendly facilities, accessible care, technology integration, stakeholder participation, and the fusion of traditional and modern care practices in the journey towards sustainable development.

Our study aligns with existing literature, emphasizing the significance of diverse social governance in elderly care through stakeholder engagement, including government policies, community participation, and private sector involvement [10,11]. These approaches improve care quality and accessibility. Sustainability principles of social equality, economic viability, and environmental sustainability are vital [5,7,8]. They align with our findings on governance efficiency. Elderly service accessibility is influenced by geographical and affordability challenges [30,31]. Our findings highlight community-driven care and technology solutions to enhance accessibility. Efficient resource use and technology integration are emphasized [38,46], enhancing care quality and efficiency. Socioeconomic factors influence elderly care [50]. Our study supports this, focusing on income levels, education, employment, social inequality, and healthcare infrastructure.

As we move forward, the insights derived from our research must serve as a bridge between existing knowledge and innovative concepts. These findings provide actionable data points that can drive real-world changes in geriatric care, guiding us toward an era of more inclusive, sustainable, and effective elderly services. In sum, our study makes a significant contribution to the discourse on geriatric care, emphasizing the need for practical action and collaboration among all stakeholders to create a better future for the elderly population. The multifaceted nature of geriatric care necessitates a multifaceted approach, and our findings serve as a beacon, guiding us toward a more compassionate and sustainable future.

**Author Contributions:** Conceptualization, H.W. and P.C.C.; Methodology, H.W. and R.Z.; Software, W.S., X.Z. and R.Z.; Validation, H.W., P.C.C. and R.Z.; Formal analysis, X.Z. and R.Z.; Investigation, H.W., P.C.C., W.S., X.Z. and R.Z.; Data curation, P.C.C.; Writing—original draft, H.W. and X.Z.; Writing—review & editing, P.C.C.; Visualization, W.S.; Project administration, W.S. and X.Z.; Funding acquisition, W.S. All authors have read and agreed to the published version of the manuscript.

**Funding:** This research received no external funding.

**Institutional Review Board Statement:** The study was conducted in accordance with the Declaration of Helsinki and approved by the Institutional Review Board (or Ethics Committee) of East China Normal University, Shanghai 200062, China (protocol code: IRB20230206, approved on 06-02-2023) for studies involving humans.

**Informed Consent Statement:** Informed consent was obtained from all subjects involved in the study.

**Data Availability Statement:** The data associated with this study is private and confidential, subject to legal and ethical restrictions. Due to the sensitive nature of the data and privacy concerns, we are unable to publicly share or disclose it. We are committed to upholding the privacy and confidentiality of the data, in accordance with legal and ethical standards. For any inquiries regarding the data used in this study, please contact the corresponding author for further details or potential access, where permissible under applicable regulations and ethical considerations.

**Conflicts of Interest:** The authors declare no conflict of interest.

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
