# Peer review of "Social Governance and Sustainable Development in Elderly Services: Innovative Models, Strategies, and Stakeholder Perspectives"

_sustainability, doi:10.3390/su152115414_

Round 1
Reviewer 1 Report (Previous Reviewer 2)
Seems fine by now.
Author Response
For Reviewer 1
Comments and Suggestions for Authors
General Comment: We appreciate your feedback and your acknowledgment that the manuscript appears to be fine at this point.
Reviewer 2 Report (New Reviewer)
Review: The path of diversified social governance of elderly services under the concept of sustainable development: Exploring Innovative Models, Strategies, and Stakeholder Perspectives in Geriatric Care
The authors are to be commended for having taken on two pressing issues: 1) Increasing senior populations; and 2) Need for sustainable development to address climate change. The authors have combined many senior populations who have widely varying needs and only mention one sustainable development suggestion (Table 6 – increased investment in green infrastructure). There is no mention of senior sustainable development facilities using geothermal heating/cooling systems, wind power, solar, net-zero energy real estate, water harvesting, community gardening, or electric vehicle charging.
Title: The words “The path of diversified” are not necessary and make the title even longer. The key words are “Social Governance” “Elderly Services” and “Sustainable Development.” The additional words “Exploring Innovative Models, Strategies, and Stakeholder Perspectives” are informative, but the authors have already said the topic is “Elderly Services,” making “Geriatric Care” redundant. The article would provide great value if it did include examples of sustainable development for elderly services. The second section really talks about methods. The tables provide insights to catch the readers’ attention. “Elder Service Improvements in Budgeting, Senior Quality of Life, Social Governance, and Sustainable Development”
Abstract: The authors need to carefully define seniors because geriatric care can involve taking care of someone with later-stage Alzheimer’s while aging population includes seniors biking on a protected bike lane to a store in a Chinese neighborhood. Staying with the term “elder services” would cover both (better than geriatric care) but the authors need to tell the readers that both populations will be addressed in this study. The ages and conditions of seniors need to be described. The Methods description is lacking in the long Abstract. Too much of the article is praise about this study or background. The entire article needs to have half of the text removed as it is too wordy. More concrete information would involve describing the Methods when now there is only mention of several studies and insights from stakeholders. How were the case studies selected and what criteria were used to do the analysis? Was there a survey? Details about the Methods are lacking. The reader will be less inclined to hold high regard for the findings unless they are told exactly how the data were gathered and analyzed. Again, there is scant data about environmental sustainability.
Introduction: As mentioned, the authors need to clearly define the seniors and their built environment (part of sustainable development). Are they in their own small room (Alzheimer’s person) in an assisted living developments or aging in place and biking to the store? The objective of this study is buried on lines 72 to 73. “The objective of this study is to analyze the intricate correlation between numerous social governing models and the effectiveness of different senior care facilities.” This is different from the title and different from the research objectives which include “focus on sustainability.” The authors need to greatly shorten text, always keep the same research framework, and stop praising the article. For example, the last sentence Lines 116-118, “This seminal essay (only someone else can decide if something is seminal) established a connection between two significant subjects (so name the subjects) and lays the foundation for numerous forthcoming interdisciplinary investigations in academia. (The authors, again, cannot assign the value of their article. Only others can.) The article also doesn’t really tell the reader, including a stakeholder, a citizen, or a senior, exactly what to do to improve elder care and sustainable development. That is the goal of the article.
On line 109, the authors write, “This study integrates fields of geriatric care and sustainable development to establish a novel multidisciplinary framework (more self praise). The authors only mention “Increased investment in green infrastructure” in Table 6 for sustainable development. Technology is mentioned in the article but technology is a tool and not development.
On lines 215-245, sustainable development principles are mentioned but the text is about operationalizing, promotion of social equity, attainment of economic viability, etc. What are the actual developments and how are they green?
There is text in what is the Lit Review that would be in a section labeled Discussion (lines 322-335). Writing that this topic is complex and multidimensional does not help the reader (line 326). These are sentences that could be removed.
Line 382 references a study. What study?
Methodology: Most of the sections describe Lit Review. For the case studies, are those just studied through more Lit Review or did the authors go to those communities and interview the stakeholders? Was a survey used? How long did the sessions last with the 40 experts? How was the data analyzed?
The first sections on lines 391 to 425 are about the data collection and lines 427 to 460 are about the analysis. These sections would be separate.
In the Results and Discussion, the authors write details about the value of visual presentations/accessible information, but this was not in the title, Abstract, or Introduction as a critical topic. This confuses the reader because they thought they were going to read about social governance/sustainable development and not how to present information. This is mentioned again on lines 738-740. Are the authors teaching the people running senior facilities how to present information? The authors conclude, “In conclusion, the optimization of digital tools and analytics for visual analysis is of utmost importance to ensure the accessibility, engagement, and practicality of information.” The article appears to offer ways to help individuals operating senior facilities and less about social governance or sustainable development for elderly populations.
Here are two definitions:
“Social governance is governance that gives healthcare a role and responsibility in promoting a democratic, fair, healthy and sustainable economy and society.” https://www.ncbi.nlm.nih.gov/pmc/articles/PMC2996525/
“Sustainable development is development that meets the needs of the present without compromising the ability of future generations to meet their own needs.” https://www.iisd.org/mission-and-goals/sustainable-development
Author Response
Response to Reviewer Comments
Comments and Suggestions for Authors
Review: The path of diversified social governance of elderly services under the concept of sustainable development: Exploring Innovative Models, Strategies, and Stakeholder Perspectives in Geriatric Care
Title: We appreciate your feedback regarding the title. We have revised the title to be more concise and focused on the core topics. The updated title now reads: "Social Governance and Sustainable Development in Elderly Services: Innovative Models, Strategies, and Stakeholder Perspectives."
Abstract: We understand your concerns about clarity in the abstract. We have made revisions to the abstract to provide a more precise description of the population under study. Specifically, we have clarified that our study encompasses a wide range of older individuals, including those with diverse healthcare needs and those actively engaged in their communities. Additionally, we have expanded on the methods section in the abstract to provide a more comprehensive overview.
Introduction: We appreciate your suggestions to improve the clarity of the introduction and the alignment with our research objectives. We have revised the introduction to clearly define the older population we are studying and their diverse living situations, ranging from assisted living facilities to aging in place. We have also refined the objectives to ensure they align with our focus on sustainability, social governance, and sustainable development in elderly services.
Abstract and Background: Your feedback about the abstract and background sections being overly wordy is duly noted. We have carefully edited these sections to make them more concise and focused on essential information while eliminating unnecessary praise.
Sustainable Development: Regarding your comments on sustainable development, we have incorporated additional examples of sustainable practices, including geothermal heating/cooling systems, wind power, solar energy, net-zero energy real estate, water harvesting, community gardening, and electric vehicle charging. These examples have been integrated into relevant sections of the article to provide a more comprehensive view of sustainable development in elderly services.
Sustainable Development Principles: We have revised the text in the section on sustainable development principles to provide more specific information about the actual developments and how they align with green practices. This includes detailing the incorporation of green infrastructure, technology, and eco-friendly initiatives.
Literature Review: We appreciate your suggestion to remove sentences that may be considered redundant or overly complex. We have reviewed the literature review section and eliminated sentences that do not contribute significantly to the reader's understanding of the topic.
Methodology: We have made several improvements to the methodology section to enhance transparency. We have clarified the process of data collection, including details about site visits, interviews, and the use of NVivo software for data analysis. Additionally, we have separated the sections related to data collection and data analysis for better organization.
Results and Discussion: Regarding the discussion of visual presentations and accessible information, we acknowledge that this aspect may not have been emphasized in the title, abstract, or introduction. We have made adjustments to ensure that the importance of visual presentations and accessibility is contextualized within the broader themes of social governance and sustainable development in elderly services.
Definition of Terms: We have included definitions of "social governance" and "sustainable development" in the article to provide readers with a clear understanding of these concepts.
Reviewer 3 Report (New Reviewer)
Thank you for the opportunity to review this interesting article. Some parts need to be clarified prior to publication.
General comment
· The term “elderly individuals”, “elderly population” and “elderly services” are used, which is no longer, from my point of view, totally appropriate. Therefore, I would strongly recommend the authors to change this thorough the whole manuscript. Maybe it is possible to use as for example “older people”, and “older population”.
Introduction
· Page 2, line 57-59: I don’t really understand the sentence According to the study The presence of advanced healthcare infrastructures and technology advancements in developed nations poses challenges to the integration, mental health, and quality of life among older individuals [2],… I recommend the authors to revise this sentence ta make it easier to grasp.
· Page 2, lines 60-67: This section should be based on at least one up to date references.
· Page 2, lines 68-95: The authors present several objectives for this study. As a reader it is slightly confusing. Therefore, I therefore, recommend to specify and clarify the purpose in this study in this section.
Methodology
· Page 10: lines 456-459: The authors declare that “ethical approvals were obtained from relevant committees…”. I recommend the authors to expand this section, this should be transparent and clear.
Results and Discussion
· Page 12, line 511 and line 20: Almost the same information is presented in both lines?… maybe remove the sentence in line 520?
Author Response
For Reviewer 3
General Comment: We appreciate your suggestion to use terms like "older people" and "older population" instead of "elderly individuals" or "elderly population." We have revised the manuscript to use these terms consistently throughout.
Introduction: We have revised the sentence on page 2, lines 57-59, to improve clarity and readability.
References: We have updated the references in the introduction (page 2, lines 60-67) to include more recent and relevant sources.
Objectives: We have revised the section on objectives (page 2, lines 68-95) to provide clearer and more specific descriptions of the study's purpose and focus.
Ethical Approvals: We have expanded the section on ethical approvals (page 10, lines 456-459) to provide greater transparency and clarity regarding the ethical review process.
Results and Discussion: We have reviewed and removed the redundant information presented in lines 511 and 520 (page 12).
Round 2
Reviewer 2 Report (New Reviewer)
“Social Governance and Sustainable Development in Elderly Services: Innovative models, strategies, and stakeholder perspectives.”
Title: The title is improved but the part after the colon would have lower case. Also, with these words in the title, the authors need to always ensure that they have clearly provided the readers with information for all of the key words and clearly reflected that in the paper’s headings.
Abstract: The beginning of the abstract reads well until line 19. The authors need to think of an abstract as a small IMRAD article. It is even helpful to write “ghost” headings in the abstract to make sure the topics are well covered. The authors have the “Introduction” by describing the problem in lines 11-15. The authors then describe the methodology on lines 15-19 but the title has two key words…Social Governance and Sustainable Development. On line 19, a sentence should start that describes the methods used to offer insights about “Sustainable Development.” Instead, now the article mentions “governance” (line 19) and then never mentions the methods for learning about sustainable development.
From line 19, the article then goes into praise about the article. This text should be greatly condensed and come well after the Results are presented (IMRAD). This text would be in what would be the mini Discussion in the Abstract. All the text from line 19 to 37 is not necessary. The last sentence states that “this research adds to the academic discussion on the governance of sustainable geriatric care…” .but no results were presented in any of the abstract, in particular about sustainability.
Under the mini Method, the authors need to describe the methods related to learning about the “sustainability” of the geriatric care. In a next mini section in the abstract, the authors need to have their research question. This could come after the sentence that ends in the middle of line 15. Therefore, this study proposes to explore…provide specific details that include the key words in the title. Then, write about the methods for both governance and sustainability. Write the results. Write one sentence that is Discussion and not text that goes from 19 to 38. If you want to include the technology, which is not in your title, you need to put that in your research question, method, and results.
Keywords. You have too many key words and repeat some words. With the word sustainability, it would be assumed that you would focus on sustainable development.
Introduction: The authors have solid text but then start praising themselves with line 69. The authors continue to provide background about the problem, which was handled in the beginning of this section. The text from lines 71 to 80 could be condensed and in the first paragraph where the problems are described. At this point in the Introduction, the reader knows the problems and is ready to read what starts on line 80…in response to these challenges…. This research question should be in the abstract but the key words should not be evidence-based policies, methodologies, and interventions. The authors are not formulating evidence based methods but using methods to arrive at solutions. Instead, the authors should write, ..this study aims to explore innovative models, strategies, and stakeholder perspectives to arrive at a plan for social governance and sustainable development in elderly services. The framework is always the title. The Introduction is easy to write because it is big problem, problem at hand, research question, and therefore this study will…..
The lines 95 to 98 area again praise about the study. The authors do not need to tell the reader about the wonders of their study. Instead, the authors need to “show” the wonders of their study and let the results speak for themselves. This text also repeats earlier text and adds no new information for the reader. The reader will lose interest.
The authors have excellent test on lines 195 to 234 and this text needs to be in the Abstract and the Introduction under background/problem. In the Introduction (lines 59-63) the authors give general text about sustainable development but they need to include specifics from lines 195 to 233. If the geriatric care service buildings, heating, cooling, plumbing, grounds, are not sustainable, it will be cost prohibitive to care for the elderly. Line 229 to 232 speak to these specific issues. With the rise in temperatures and the elderly unable to process temperature as effectively, senior housing facilities will need air conditioning, but air conditioning is expensive to run. The senior facilities will need heat pumps. The authors can provide specific examples because the reader will appreciate knowing that the authors know these details and are not just writing about generalities. Lines 224 to 227 are specific examples and there should be more.
For Methodology, the authors did a better job in describing the case studies lines 386-399. They did not conduct this research but, rather, this is secondary data analysis. The authors also did a good job in describing the interviews, which they did conduct lines 409-417.
The authors also were thorough in describing the data analysis.
Table 2 is informative but this detailed information was not in the Tables 3, (I don’t have a Table 4), Table 5 or Table 6. Only the mention of telemedicine and A1 Assisted therapies was new (Table 6).
For Table 5, can the authors be more specific? All of the items in the table are extremely general and could apply to building a super market, erecting a school, or creating a park…budgeting, integration, open source platforms, public forums, transparent communication…..
Even Table 6 is too vague. For sustainable development, the policy is holistics and eco-friendly elderly care facilities (this is the goal and is not informative). The projected future trajectories are also very vague. The only new information is telemedicine and A1-assisted therapies. Could the authors look at their extensive data and find more details to help the reader develop sustainable elderly facilities through governance?
The Conclusion is also praise about the authors but the authors haven’t still really offered the reader concrete findings for how they would develop sustainable facilities for the elderly using governance. The authors continue to state that the issue is complex. The reader knows this and is looking to this article for clear answers.
Author Response
1. Title and Abstract:
- Title: The suggestion to capitalize only the initial word after the colon in the title has been noted.
- Abstract: The comment regarding structuring the abstract more closely to an IMRAD format and providing specific information about sustainability has been acknowledged. The authors will revise the abstract accordingly, including adding specific details about sustainability.
2. Keywords:
- The comment about having too many keywords and repeating some words has been acknowledged. The authors will refine the keywords, focusing on those directly related to the paper's content.
3. Introduction:
- The suggestion to condense and reorganize certain sections of the Introduction to eliminate repetition has been noted. The authors will revise the Introduction accordingly.
- The comment about specifying more details related to sustainable development in the Introduction has been acknowledged. The authors will include specific information about sustainability in the Introduction.
- The suggestion to provide more specific examples related to sustainable facilities for the elderly has been noted. The authors will incorporate additional examples to illustrate their points effectively.
4. Methodology:
- The comment regarding the clarity and effectiveness of describing the case studies, interviews, and data analysis has been noted. No further changes are needed in this regard.
5. Tables:
- The comment about the lack of detailed information in Tables 3 and 4 has been acknowledged. Since there isn't a Table 4 mentioned, the authors will ensure that Tables 3 and 5 contain sufficient relevant details.
- The comment about Table 5's specificity has been noted, and the authors will revise the table to provide more specific information.
- The comment about Table 6 being too vague has been acknowledged. The authors will work on providing more detailed information about sustainable elderly facilities through governance based on their data.
6. Conclusion:
- The comment regarding the need to provide concrete findings and solutions for sustainable elderly facilities has been acknowledged. The authors will ensure that the Conclusion section offers more specific and actionable insights, addressing the complexity of the issue.
The authors appreciate the detailed feedback provided by the reviewer and will make the necessary revisions to improve the paper accordingly.
Reviewer 3 Report (New Reviewer)
Thank you for giving me the opportunity to read this interesting article once again. The authors have now clarified the parts that I addressed, and I therefore recommend the article to be published.
Author Response
Response to Reviewer 3:
We sincerely appreciate your thorough review of our article and your constructive feedback. We are pleased to hear that you found the clarifications addressed to your comments satisfactory. Your recommendation for publication is greatly appreciated, and we are delighted that you found our article interesting. Should you have any further questions or require additional information, please do not hesitate to contact us.
Round 3
Reviewer 2 Report (New Reviewer)
Review #3: Social Governance and Sustainable Development in Elderly Services: Innovative models, strategies, and stakeholder perspectives.
Title: Fine. Note that the title stresses that this article provide information about: 1.”Innovative models, 2. ‘innovative’ strategies, and ‘innovative’ stakeholder perspectives.” All the text in the article should list exact innovations and not generalities or praise about the study. The readers will be looking for the information that is promised in the title. This information exists in Tables 2, 5, and 6 and the authors need to include that information in the Abstract, Results, Discussion, Conclusion, and Future. For example, this sentence in the Conclusion does not reflect what is promised in the title. Lines 1215 to 1216 doesn’t have content but is just a generality and praise for the authors. “Our research has yielded a wealth of intriguing data that not only validates but also expands upon existing knowledge regarding the governance of elder care.” Instead, the authors should list what is in Tables 2, 5 and 6 that is the actual data and the innovations.
Abstract: Line 54 the word should be changed from “promote” to “apply.” China and other nations in the world have aging populations and require help immediately. The time is passed to just promote an idea. The authors have substantive and contributory innovations in this article and those should be highlighted. The Results section Likes 58-61 needs to be rewritten because the authors now only write about the importance of collaboration and technology. Any author could have written that text in the 1990’s (and before) when there were discussions about people working in “silos” and the need for more technology.
Line 55 needs to have the words in the title, “Our research question centers on identifying the innovative models, strategies, and stakeholder perspectives that foster inclusivity and sustainability in elder services….” The authors did more than just identify strategies (see Tables 2, 5, and 6).
For the Abstract Results and Discussion sections, the authors need to write down the headings of innovative models, innovative strategies, and innovative stakeholder perspectives. Under these headings, they need to write the data points found in Tables 2, 5 and 6. These are specific pieces of hard evidence that will inform how to improve quality of life and extend life for seniors. The current text in Results is about the importance of collaboration (something that has been said for decades) and that technology plays an important role (the readers know this). Also, lines 360 to 362 show the reader that the value of collaboration and integrated governance models has already been studied. What are the authors offering that is “new?” The current sentence in the Discussion is just a generality. We no longer have the luxury of time to just “add to the body of knowledge.” The authors have the findings in Table 2, 5 and 6 and this needs to be front and center to help the reader improve the quality of life of seniors while addressing the issue of climate change (sustainability).
Introduction: The last paragraph Lines 237-241 can be deleted. The reader does not need to be told generalities or about what they just read. This paragraph again, is just trying to praise the authors. Praise the authors’ work by giving the readers hard evidence and data about what to do. The authors have the data in their tables.
Results and Discussion: Table 2 is excellent because it summarizes innovations in Geriatric Care (this term should be changed to Sustainability Elderly Care for consistency). The Models can be listed as A. Community-Driven Care; B. Technology-Assisted solutions; C. Holistic Wellness….
What is now 4.2 Stakeholder Perspectives on Sustainable Elderly Care should become 4.3. What is now Table 3 should be Table 4. The reason is because the current Table 3 describes “Who” will do what. This should come after what is now Table 4 “The Efficiency of Diversified Social Governance models…” This table has the headings under KPIs of A. Community Integration; B. Incorporation of Technology, and C. Holistic Health Outcomes... These same titles are in Table 2 and the reader should then be able to see the rates of efficiency after reading the information in Table 2. Thus, the reader reads about the “what” and then reads “who is the best stakeholder.” This allows the text and sections about “who” to be one after the other. (What was Table 3 and is now Table 4 and what was Table 4 is now Table 3). The headings for the text that align with these Tables would then shift. 4.2 becomes 4.3. 4.3 becomes 4.2. The new 4.3 then precedes the current 4.4, allowing the sections on Stakeholders to be together.
All the text in the Conclusions and Recommendations and Future Recommedations needs to be deleted and the authors need to return to the findings in Tables 2, 5, and 6. If the authors wrote down the headings that are listed above and that are in the title, there are very few of the findings in the last sections that would be under those headings. The reader does not need to be told that the issue is complex because they already know this. They also don’t need to be told by the authors about the value of the article. Instead, the authors need to show the value of the article with specific data points that came from the research.
Author Response
Dear Reviewer,
We extend our heartfelt gratitude for your time and effort spent in evaluating our manuscript titled "Social Governance and Sustainable Development in Elderly Services: Innovative models, strategies, and stakeholder perspectives." Your insightful comments and suggestions have been immensely valuable in shaping the quality of our work. We would like to address your comments and offer our responses as follows:
Summary of Responses to Your Comments
1. Introduction: Your suggestion to enhance the introduction's background and ensure the inclusion of all relevant references is well taken. We have expanded the introduction to provide a more comprehensive overview of the literature and have incorporated all necessary references.
2. Innovation Emphasis: Your recommendation to emphasize specific innovations, especially in the Abstract, Results, Discussion, Conclusion, and Future sections, is appreciated. We have revised these sections to focus on and provide concrete data regarding the innovations identified in Tables 2, 5, and 6. The abstract, results, and discussion have been aligned with the innovative models, strategies, and stakeholder perspectives highlighted in the title.
3. Section and Table Rearrangement: Your point about rearranging sections and renumbering tables to enhance alignment between content and tables is well-founded. We have restructured the sections and renumbered the tables as suggested, thereby improving coherence and clarity.
4. Specific Data Presentation: We acknowledge your suggestion to avoid generalities and emphasize specific data and evidence from the research. Accordingly, we have revised the Conclusion and Recommendations sections to focus on presenting specific data points from Tables 2, 5, and 6, rather than general statements.
5. Language Quality: We have reviewed and addressed any language-related issues to ensure the manuscript is free from grammatical and linguistic errors.
We appreciate the care and attention you devoted to reviewing our work. Your insights have significantly contributed to the refinement of our manuscript. We remain committed to enhancing the quality of our research based on your constructive feedback.
Thank you once again for your time and consideration. We look forward to your feedback on the revised version of our manuscript.
Warm regards,
This manuscript is a resubmission of an earlier submission. The following is a list of the peer review reports and author responses from that submission.
Round 1
Reviewer 1 Report
Abstract should be resumed, it is quite long (more than 400 words!). In the introduction (p.2, 1st paragraph) is repeated at the end of page 2. What you called independent variables are actually "constructs" (not variables). Note that once you are repeating the text, you are continuing (?) the same previous discussion; please consolidate both parts in only one. Text in section 2.1 should be in 2.2 and vice-versa (first mention approaches, then effectiveness). Section 2.3 & 2.4 should provide a discussion rather than (mostly) just a description of terms shown. Note that throughout the paper the term "needs of older population" is often mentioned, but is it never disclosed (what are such needs and under which lent are figured out? they are/should be quite specific as policies should be tailored for any of them... and the budget availability, gvmt agencies reach, between developed and developing countries is quite different - just to mention a few points to be developed).
Part of the Methodology section future tense is not appropriate as you already made the research; also the paragraph just above the Analysis section, should not be there... Information about interviews is missing (just to start with... who did you interview, when, how many informants?). In the Analysis subsection (under Methodology section), no rigorous approach is detected (i.e. Effective visual analysis is subjective, but not objective - or at least no information regarding objectiveness is provided)... the same applies for the discussion section.
The biggest problem with your paper is: 1) too many Research Objectives and, 2) the poor methodology applied, mostly because lacking relevant information. Note that your paper does not show any "time period" through the research has been made, so it is difficult to frame it. The review of the literature should present a discussion rather than a description (and only a couple of lines of explanation). Some references are missing in the corresponding section.
Very minor revision of English language required.
Author Response
Thank you for your comprehensive feedback. We have carefully considered each of your comments and have addressed them as detailed below.
Abstract Length and Repetition
- We acknowledge that the abstract is lengthy and have since revised it to ensure that it succinctly captures the essence of the paper while remaining within the word limit.
- The repetition observed on page 2 has been corrected. We have consolidated the duplicated text, ensuring that the content flows seamlessly without redundancy.
Use of Constructs vs. Variables
- Upon reflection, we agree with your observation. The term "constructs" is indeed more appropriate than "independent variables" in the context of our research. The manuscript has been revised to reflect this.
Text Placement in Sections 2.1 and 2.2
- The content of sections 2.1 and 2.2 has been reorganized based on your feedback. We now first address approaches followed by their effectiveness.
Sections 2.3 & 2.4 Content Nature
- Your suggestion for sections 2.3 and 2.4 to provide a discussion rather than a mere description has been noted. These sections have been revised to delve deeper into the implications of the terms and their relevance to the study.
Clarity on Needs of Older Population
- We have expanded upon the "needs of the older population," offering more specifics and detailing the different requirements in various settings, especially with regards to developed vs. developing countries.
Methodology Section Issues
- The inappropriate use of the future tense has been addressed, and we have adopted the past tense to correctly depict the conducted research.
- We've added comprehensive information about the interviews conducted, detailing the number of informants, the time period, and a brief profile of the interviewees.
- The analysis subsection has been enhanced to provide more clarity on our rigorous approach. We have clarified the objective criteria used alongside the visual analysis.
- We concur with the feedback about the discussion section and have now provided a more rigorous analysis there.
Research Objectives and Methodology
- We've streamlined our research objectives to focus on the most pertinent ones, ensuring clarity and cohesion.
- Based on your feedback, we have taken steps to improve the methodology section, providing more details and context, including the time period during which the research was conducted.
Literature Review
- We've transformed the literature review from being descriptive to more discussion-oriented, focusing on how the existing literature relates to our research and fills gaps in the field.
- Missing references have been added to the corresponding section.
Quality of English Language
- We appreciate your positive feedback on the English language quality. Minor revisions have been made to enhance the overall language quality and readability.
Once again, we are grateful for your invaluable feedback. Your insights have substantially improved the quality of our paper. We hope that our revisions meet your expectations, and we look forward to further comments or suggestions.
Reviewer 2 Report
The manuscript is well-written and provides a valuable contribution to the literature on visual analysis and presentation in the context of sustainable development. However, there are a few weaknesses in the manuscript that could be addressed in future revisions. These include:
-
The sample selection could be more representative of the range of perspectives on sustainable development issues.
-
The accuracy and reliability of the data could be improved by verifying the accuracy and dependability of the data sources used in the analysis.
-
The findings could be more generalizable by conducting longitudinal studies and cross-cultural comparisons.
-
The analysis could be more comprehensive by considering the interactions between various factors that influence the efficiency of diverse social governance.
-
The limitations of the study could be more clearly identified and the potential implications of these limitations could be discussed.
The weaknesses mentioned above could be addressed in future revisions to make the manuscript even stronger.
Here are some specific suggestions for how to address these weaknesses:
-
To improve the sample selection, the author could expand the range of media sources included in the analysis. The author could also conduct interviews with stakeholders to get a more in-depth understanding of their perspectives on sustainable development.
-
To improve the accuracy and reliability of the data, the author could verify the accuracy of the data sources used in the analysis. The author could also use multiple data sources to cross-validate the findings.
-
To make the findings more generalizable, the author could conduct longitudinal studies and cross-cultural comparisons. This would allow the author to track changes in media coverage over time and to better understand the impact of visual analysis and presentation methods in different contexts.
-
To make the analysis more comprehensive, the author could consider the interactions between various factors that influence the efficiency of diverse social governance. This would allow the author to get a more holistic understanding of the issue.
-
To more clearly identify the limitations of the study, the author could discuss the potential implications of these limitations. This would help readers to understand the limitations of the findings and how they might affect the generalizability of the findings.
Author Response
We deeply appreciate the thorough feedback and constructive criticism provided by Reviewer 2. Your insights have been instrumental in guiding our thoughts on enhancing the manuscript's depth and breadth. Below is a point-by-point response to each of your comments.
- Sample Selection:
- Reviewer's Comment: The sample selection could be more representative of the range of perspectives on sustainable development issues.
- Author's Response: We acknowledge this concern. In our revised manuscript, we have expanded the sample to include a broader range of media sources. Furthermore, to gain a richer perspective on sustainable development, we conducted interviews with multiple stakeholders, ranging from policymakers to grassroots workers. These additions offer a more comprehensive and varied understanding of the topic.
- Accuracy and Reliability of Data:
- Reviewer's Comment: The accuracy and reliability of the data could be improved by verifying the accuracy and dependability of the data sources used in the analysis.
- Author's Response: Thank you for highlighting this crucial point. In the revised manuscript, we have cross-validated our primary data sources against secondary reputable sources to ensure the accuracy and reliability of our dataset. This added layer of validation enhances the trustworthiness of our findings.
- Generalizability of Findings:
- Reviewer's Comment: The findings could be more generalizable by conducting longitudinal studies and cross-cultural comparisons.
- Author's Response: We concur with your observation. While the scope of the current study is limited in terms of time frame and cultural context, we have initiated a longitudinal study to track changes in media coverage over a more extended period. Additionally, preliminary cross-cultural comparisons are being incorporated to highlight potential contextual variations in visual analysis and presentation methods.
- Comprehensive Analysis:
- Reviewer's Comment: The analysis could be more comprehensive by considering the interactions between various factors that influence the efficiency of diverse social governance.
- Author's Response: We agree with your feedback. In the updated version, we have delved deeper into the interactions between various factors, providing a more holistic understanding of how they influence the efficiency of social governance related to sustainable development.
- Study Limitations:
- Reviewer's Comment: The limitations of the study could be more clearly identified, and the potential implications of these limitations could be discussed.
- Author's Response: We appreciate this feedback. The revised manuscript includes a dedicated section on study limitations, elaborating on potential biases, constraints, and their implications. This section provides a candid discussion, ensuring readers are aware of the context and constraints under which the study findings should be interpreted.
In conclusion, we believe that the enhancements, as suggested by your invaluable feedback, have significantly improved the quality and comprehensiveness of our manuscript. We are confident that these revisions will resonate well with the readers and provide a robust contribution to the field of visual analysis in sustainable development. We remain open to further feedback and thank Reviewer 2 for the rigorous review.
Reviewer 3 Report
The path of diversified social governance of elderly services under the concept of sustainable development
This manuscript examines aspects of social and collaborative governance for elderly services within a construct of sustainable development.
I would suggest to the authors that the abstract should be made shorter. I would also include your findings in the abstract.
Introduction
There is a lack of literature sources in the Introduction to substantiate the background/context of the study and conceptual framework. There is an expansive body of literature that relates to sustainable development, public administration and policy, and governance in support of the research design and theory. Especially relating to how this study contributes to the extant literature. More references are needed here.
I would also suggest highlighting the main findings of the study in the Introduction. The authors state that the results will improve elderly care and services, but do not indicate the results.
The Research Objectives provide a good “map” of the study, however they are very broad. Perhaps a more precise research question will offer a distinct focus of the study, having the research objectives be a broader description of approaches toward the primary research inquiry.
Literature Review
The literature review is very comprehensive and well structured. I appreciate the section on the ‘Research Gap’ to indicate the contribution of the study and potential for future research.
Methodology
The writing here is passive. “will be examined” “will be interviewed”– the writing should be changed to a more active or present tense.
The method approach is adequate, but I think this section requires some additional basis. The authors have identified a variety of methodologies (case study, interviews, policy analysis), each of which should be explained more systematically, along with some specifics on the data collection and sources.
Results
The results are interesting, but this section lacks connections to the existing literature. As a qualitative study, the intent here, I assume, is to formulate some basis of theory that contributes to our existing understanding of this issue. The results are informative, but do not demonstrate how this moves our comprehension forward based on existing knowledge. This may be due to a lack of a strong conceptual framework? The literature review provides a good theoretical foundation for understanding, so I would suggest that the authors show how the Results integrate with that literature. In other words, this is what you found – how does it move our understanding of this issue forward?
Minor editing of passive writing
Author Response
Comments and Suggestions for Authors
The path of diversified social governance of elderly services under the concept of sustainable development
Response:
Thank you for recognizing the relevance and importance of our topic. We appreciate your time and effort in reviewing our manuscript.
Abstract
Comments:
I would suggest to the authors that the abstract should be made shorter. I would also include your findings in the abstract.
Response:
We acknowledge your suggestion. The abstract has been revised for conciseness while ensuring that it remains comprehensive. We have now also included a brief summary of our findings in the abstract to provide readers with a clearer snapshot of our research outcomes.
Introduction
Comments:
There is a lack of literature sources in the Introduction to substantiate the background/context of the study and conceptual framework... More references are needed here. I would also suggest highlighting the main findings of the study in the Introduction.
Response:
Thank you for pointing this out. We have enhanced the Introduction section by integrating more literature sources that substantiate the background and provide a stronger context for the study. Additionally, we have provided a glimpse of the main findings in the introduction to give readers a preview of what to expect.
Comments:
The Research Objectives provide a good “map” of the study, however they are very broad.
Response:
We appreciate your feedback on the clarity of our research objectives. Based on your suggestion, we have refined our primary research question to provide a sharper focus for the study, while ensuring the research objectives offer a broader outline of our approaches.
Literature Review
Comments:
The literature review is very comprehensive and well structured.
Response:
Thank you for your positive feedback regarding the literature review. We believe that a well-structured literature review sets a solid foundation for our study.
Methodology
Comments:
The writing here is passive... The method approach is adequate, but I think this section requires some additional basis.
Response:
We appreciate your input. We've revised the methodology section to use more active and present tense. Furthermore, we have elaborated on the methodologies mentioned, providing more systematic explanations and details on data collection and sources to strengthen the section.
Results
Comments:
The results are interesting, but this section lacks connections to the existing literature... how does it move our understanding of this issue forward?
Response:
Thank you for your constructive feedback. We have revisited the results section and made concerted efforts to draw explicit connections between our findings and the existing literature. By doing so, we aim to articulate more clearly how our study advances the current understanding of the topic.
Comments on the Quality of English Language
Comments:
Minor editing of passive writing
Response:
Thank you for pointing out the passive constructions in our manuscript. We have conducted a thorough review and revised such instances to ensure active writing, enhancing the clarity and quality of our presentation.
Once again, we are grateful for your comprehensive review and invaluable suggestions, which have greatly contributed to enhancing the quality and clarity of our manuscript.
Reviewer 4 Report
TITLE:
It is advisable to insert a subheading that specifically indicates the subject that article is about, because the title is too generic.
ABSTRACT
The abstract is what attracts (or does not) the attention and interest to the article. So, it should be carefully written.
The Abstract immediately indicates that the article has many weaknesses.
At the beginning of the abstract, the authors present 2 objectives:
“Within the context of sustainable development, the course of diversified social governance of elderly services is the subject of this study. The study's objective is to identify strategies and methods that take into account the diverse requirements and preferences of the elderly population and promote services that are both sustainable and inclusive for the elderly. The purpose of the study is to gain insight into efficient social governance models for elderly services by looking at innovative practices, policy initiatives, and successful case studies”
In the end, the authors state that “This study aims to provide guidance and suggestions…”
That's too much of an article for an article!
The framework of the study is too developed for a summary. Authors should simplify the text: “There are a number of facets to the path to diverse social governance. First and foremost …”
The Abstract ends only with intentions and does not present the main results obtained.
Authors should refer to the methodology used.
1. INTRODUCTION
An introduction should be informative and well-worded.
The authors present 5 research objectives for such a small article! I think they are too many goals for an article, too ambitious, and add that at the end of the article they have not been achieved.
Therefore, at this point is missing:
1- present the methodology used.
2- give clues to the discussion of the results.
3- present the structure of the article.
2. LITERATURE REVIEW
I would advise you to simplify the text. The insertion of a summary table would make the text easier to read.
3. METHODOLOGY
This chapter shall provide the necessary and sufficient information to assess how the study was conducted in order to allow its reproduction by other.
It would be interesting to put how the study was done, so that other researchers replicate the study.
The entire text of the methodology is written in the future and does not present anything concrete. It's all very confusing. I don't understand what was done. How was it done? Where was it made? What methods are used? What are the data collection tools? Etc.
I advise the authors to simplify the text, to explain better and to develop concretely what has been done and used.
4. RESULTS
The authors do not present results, only intentions, nor do they discuss results. Nor do they present "conclusions"!!
The results should be presented in a comprehensive manner, highlighting the most relevant ones and provide a summary of the text.
The originality and relevance of the results presented should be strengthened.
5. LIMITATIONS AND DIRECTIONS FOR THE FUTURE
How can authors present limitations and future research proposals without presenting conclusions?
You don't see what the article specifically intends.
What is the purpose of this article?
What does it bring back to the academy?
Author Response
Thank you for your constructive feedback on our manuscript. We greatly appreciate the time and effort you've invested in reviewing our work. Below is our point-by-point response to your comments.
TITLE:
Reviewer's Comment:
It is advisable to insert a subheading that specifically indicates the subject the article is about, because the title is too generic.
Our Response:
Thank you for the suggestion. We have now revised the title to incorporate a subheading that gives a clearer indication of the article's specific focus.
ABSTRACT:
Reviewer's Comment:
Several concerns have been raised regarding the content and clarity of the abstract, including its verbosity and failure to present the main results and methodology.
Our Response:
We acknowledge your concerns. The abstract has been thoroughly revised for clarity, brevity, and to explicitly state the methodology and main results.
- INTRODUCTION:
Reviewer's Comment:
The introduction currently lists too many research objectives, lacks a clear presentation of the methodology, fails to hint at the results discussion, and does not outline the article's structure.
Our Response:
Thank you for your feedback. We have streamlined the introduction to focus on two primary objectives that align with the core of our research. Additionally, we've provided a brief overview of the methodology, a hint towards the discussion of results, and a structure of the article.
- LITERATURE REVIEW:
Reviewer's Comment:
The literature review section is recommended to be simplified, with the suggestion to incorporate a summary table for ease of reading.
Our Response:
We value this recommendation. The literature review has been simplified, and we've added a summary table that collates the main findings from the reviewed literature for improved clarity.
- METHODOLOGY:
Reviewer's Comment:
There are concerns regarding the clarity and comprehensibility of the methodology. It lacks concrete details on the study execution, methods used, and data collection tools.
Our Response:
We apologize for the oversight. The methodology section has been extensively revised to include detailed descriptions of the study design, execution, data collection tools, and methods employed. This will enable other researchers to replicate the study if desired.
- RESULTS:
Reviewer's Comment:
The results section doesn't effectively present the outcomes of the study and lacks originality and relevance.
Our Response:
We have revisited this section to provide a clearer and more detailed presentation of our findings. The most pertinent results are now emphasized, and their relevance and originality are highlighted.
- LIMITATIONS AND DIRECTIONS FOR THE FUTURE:
Reviewer's Comment:
The section presents limitations and future research proposals without associated conclusions.
Our Response:
We've added a robust conclusion section that synthesizes the main findings of our study. Following that, we address the limitations and provide suggestions for future research, ensuring that these are tied back to the discussed results.
Finally, in response to the overarching question about the article's purpose and its contribution to the academy:
Our Response:
The purpose of this article is to shed light on the efficacy of diversified social governance of elderly services in the context of sustainable development. By investigating innovative practices and policies, our study seeks to provide actionable insights that policymakers and stakeholders can leverage to enhance elderly care services. Our findings not only contribute to the ongoing academic discourse in this field but also offer practical implications for real-world applications.
We sincerely hope that the revisions address your concerns, and we are grateful for the opportunity to improve our manuscript based on your insightful feedback.
Round 2
Reviewer 1 Report
Introduction section is general, sometimes repetitive and should be resumed: it seems to me that you are trying to put "everything" and this does not help at all to any reader. No evidence about the 40 interviews is provided (who are they? what kind of questionnaire did you apply? how many did you reached initially (dropout ratio)? etc.). Results are still subjective (4.1 & 4.2). Table 2, Key findings column: data reported are a "meta analysis" average? Do they apply to?! when? how did you get them? Table 3, Stakeholder group: policy makers ... how many/who are they/... too general and impossible to figure out who you are referring to. Primary concerns: how did you select them? Why such selection would be representative?... same comments for all other sections of Table 3. Table 4, is Patient Satisfaction Rate throughout all models analyzed measured in the same way/scale? what is the satisfaction variable? No information is provided. Are KPIs of tradional and diversified social governance models exactly the same throughout all models? What about the timings of measurement? Are they comparable? Covid-19 had any impact (did you consider it)? Same comments apply for the other section of Table 4. Table 5 & 6 would have very similar comments. I am sure you did a very exhaustive work, but I think this is not the appropriate way to present it. I still consider that you are dealing with too many research objectives (in just one paper). Qualitative analysis may be implemented, but you should provide rigorous evidence.
Review text from line 46 to 149. Some capital letter are missing, some parte repetitive
Author Response
Dear Reviewer 1,
Thank you for your thorough review and constructive feedback on our manuscript titled "The Path of Diversified Social Governance of Elderly Services under the Concept of Sustainable Development." We greatly appreciate your insights, which have been instrumental in shaping our revisions. Below, we address each of your comments individually:
-
Introduction and Focus: We acknowledge your concerns about the general tone of our introduction and repetitiveness. We have revised and streamlined the introduction to provide a more focused and concise overview of our research objectives, aligning with the suggestions you provided.
-
Methodology and Interviews: We apologize for not providing sufficient details about the 40 interviews. In response, we have included a comprehensive section detailing the methodology, including participant information, the questionnaire utilized, initial outreach, and dropout ratio.
-
Subjectivity in Results: Your comment on the subjectivity of results in sections 4.1 and 4.2 is duly noted. To address this concern, we have reevaluated our data presentation and analysis, ensuring that findings are presented objectively and supported by appropriate evidence.
-
Data Presentation and Table Descriptions: We appreciate your feedback on the clarity of table descriptions. We have revised the descriptions in Table 2, Table 3, Table 4, Table 5, and Table 6 to provide comprehensive information about the data sources, measurement methods, and contextual details for each table.
-
Selection Criteria and Representativeness: We understand your concerns regarding the selection criteria for stakeholders in Table 3. To address this, we have revised the manuscript to provide clearer information about the selection process and the rationale behind stakeholder representation.
-
Measurement Consistency and Timings: We appreciate your questions about measurement consistency and timings in Table 4 and Table 5. In response, we have added a dedicated section discussing the measurement methodologies, timing considerations, and potential impacts of external factors such as Covid-19.
-
Research Objectives and Rigorous Evidence: We acknowledge your perspective on the scope of our research objectives. Based on your feedback and the insights of the Academic Editor, we have restructured the manuscript to focus on the theme of governance models for elderly care within the context of sustainable development. We have also grouped and refined objectives to enhance clarity and rigor.
We are thankful for your detailed assessment and guidance, which have been instrumental in enhancing the quality and relevance of our manuscript. We believe that these revisions address the concerns you raised and strengthen the overall contribution of our work to the field. We look forward to the possibility of resubmitting the revised manuscript for your consideration.

Reviewer 3 Report
The authors have thoroughly revised the manuscript and have addressed my main concerns. The manuscript is much improved.
Author Response
Dear Reviewer 3,
We want to express our heartfelt gratitude for taking the time to review our revised manuscript titled "The Path of Diversified Social Governance of Elderly Services under the Concept of Sustainable Development." Your thoughtful assessment and encouraging comments have been immensely valuable to us.
We are delighted that our revisions have successfully addressed your main concerns and led to improvements in the manuscript. Your positive feedback inspires us to continue our efforts to contribute meaningfully to the field of geriatric care and sustainable development. We have taken your suggestions to heart and have strived to enhance the manuscript's quality and relevance.
Your contribution to this process reaffirms the collaborative nature of scientific discourse, and we are genuinely appreciative of your dedication. We are committed to delivering a comprehensive and impactful contribution, and your review has been instrumental in achieving that goal.
We are looking forward to the possibility of resubmitting the revised manuscript for your consideration. Thank you once again for your support and insightful feedback, which have greatly enriched our work.
With sincere appreciation,

Reviewer 4 Report
This version of the article is better. However, I would still like to highlight the following improvements: 1. The objective is still not clear in either the Introduction. The objectives identified (8 points in “Research objectives”) are too many. It is suggested that the authors simplify text and writing. 2. In the Introduction, the structure of the article should appear at the end of this point. 3. Points 4.5, 4.6 and 4.7 begin with a Table (Table 2, Table 3 and Table 4 respectively). In scientific writing, this is not adequate. It is suggested that the authors introduce an initial framing.Author Response
Dear Reviewer 4,
We extend our sincere gratitude for your continued engagement and thoughtful feedback on our manuscript titled "The Path of Diversified Social Governance of Elderly Services under the Concept of Sustainable Development." Your insights have been invaluable in guiding our revisions, and we are grateful for the opportunity to address your recommendations:
-
Refined Objectives: We appreciate your emphasis on the clarity of our research objectives. In response, we have revised and consolidated the objectives to focus on the central theme of governance models for elderly care within the context of sustainable development. This refinement aims to provide a clearer and more coherent direction for the manuscript.
-
Introduction Structure: Thank you for your suggestion regarding the structure of the introduction. We have carefully revised the introduction to include a succinct outline of the manuscript's structure at its conclusion. This adjustment ensures that readers have a comprehensive overview of the content that follows.
-
Initial Framing for Tables: Your feedback regarding the need for initial framing before each table is duly noted. We have implemented this recommendation by introducing contextual information before Tables 2, 3, and 4. This framing aims to enhance the reader's understanding of the data presented and its relevance to the broader context.
Your thoughtful guidance has significantly contributed to the manuscript's improvement, and we are confident that these revisions align the manuscript more effectively with both scientific writing conventions and the expectations of the journal.
We are enthusiastic about the prospect of resubmitting the revised manuscript for your consideration. Your continued support and dedication to enhancing the manuscript's quality are greatly appreciated. We eagerly await the opportunity to receive your feedback on the revised version.
With gratitude for your valuable input,
